# Coronary plaque composition influences biomechanical stress and predicts plaque rupture in a morpho-mechanic OCT analysis

Andrea Milzi[1†]*, Enrico Domenico Lemma[2†], Rosalia Dettori[1†], Kathrin Burgmaier[3], Nikolaus Marx[1], Sebastian Reith[1]*, Mathias Burgmaier[1]*

[1]Department of Cardiology, University Hospital of the RWTH Aachen, Aachen, Germany; [2]Zoological Institute, Department of Cell- and Neurobiology, Karlsruhe Institute of Technology (KIT), Karlsruhe, Germany; [3]Department of Pediatrics, University Hospital of Cologne, Cologne, Germany

**Abstract** Plaque rupture occurs if stress within coronary lesions exceeds the protection exerted by the fibrous cap overlying the necrotic lipid core. However, very little is known about the biomechanical stress exerting this disrupting force. Employing optical coherence tomography (OCT), we generated plaque models and performed finite-element analysis to simulate stress distributions within the vessel wall in 10 ruptured and 10 non-ruptured lesions. In ruptured lesions, maximal stress within fibrous cap (peak cap stress [PCS]: 174 ± 67 vs. 52 ± 42 kPa, p<0.001) and vessel wall (maximal plaque stress [MPS]: 399 ± 233 vs. 90 ± 95 kPa, p=0.001) were significantly higher compared to non-ruptured plaques. Ruptures arose in the immediate proximity of maximal stress concentrations (angular distances: 21.8 ± 30.3° for PCS vs. 20.7 ± 23.7° for MPS); stress concentrations excellently predicted plaque rupture (area under the curve: 0.940 for PCS, 0.950 for MPS). This prediction of plaque rupture was superior to established vulnerability features such as fibrous cap thickness or macrophage infiltration. In conclusion, OCT-based finite-element analysis effectively assesses plaque biomechanics, which in turn predicts plaque rupture in patients. This highlights the importance of morpho-mechanic analysis assessing the disrupting effects of plaque stress.

*For correspondence:
amilzi@ukaachen.de (AM);
sreith@ukaachen.de (SR);
mburgmaier@ukaachen.de (MB)

[†]These authors contributed equally to this work

Competing interests: The authors declare that no competing interests exist.

## Introduction

Coronary artery disease (CAD) is one of the major causes of morbidity and mortality in the Western World (*Center for Disease Control, 2013*). One of the main challenges in modern therapy of patients with CAD is the detection of atherosclerotic plaques that do not yet limit flow in the coronary artery, but may potentially cause an acute coronary syndrome by rupturing with subsequent acute occlusion of the artery. Decades ago, these lesion entities have been defined as 'vulnerable plaques' (*Virmani et al., 2003*). Pathology-based studies (*Virmani et al., 2003*; *Falk, 1989*; *Burke et al., 1997*; *Virmani et al., 2000*) as well as intravascular imaging (*Kato et al., 2012*; *Kubo et al., 2013*; *Reith et al., 2014*; *Uemura et al., 2012*; *Ehara et al., 2004*) have been used over time to explore potential features of plaque vulnerability, that is morphological characteristics which may predispose a plaque to rupture. The presently accepted features assessed using intravascular imaging include the thickness of the fibrous cap (FCT) (*Kato et al., 2012*; *Kubo et al., 2013*; *Reith et al., 2014*; *Uemura et al., 2012*), the extent of the necrotic lipid core (*Kato et al., 2012*), the presence of macrophages (*Kato et al., 2012*; *Reith et al., 2014*; *Uemura et al., 2012*), microvessels (*Kato et al., 2012*; *Uemura et al., 2012*), small calcifications (*Ehara et al., 2004*;

**eLife digest** Heart attacks are caused by a blockage in arteries that supply oxygen to the heart. This often happens when fatty deposits (or 'plaques') that line blood vessels break off and create a clot. To identify individuals most at risk of this occurring, physicians currently use symptoms, family history, blood tests, imaging and surgical procedures. But better methods are needed.

Imaging blockages in the arteries of individuals who died from heart attacks highlighted certain plaque characteristics that increase the risk of a rupture. Further understanding the forces that lead to these fatty deposits breaking off may help scientists to develop improved heart attack prediction methods.

Using patient-specific computer simulations, Milzi et al. show it is possible to predict where plaques are most likely to rupture in an individual, based on biomechanical stresses on the deposits in the artery. The models also showed how forces on the external layers of the plaque played a pivotal role in breakages.

More research is needed to confirm the results of this study and to develop automated ways for measuring the stress exerted on plaques in the arteries. If that research is successful, biomechanical analyses of artery plaques in routine patient assessments may one day allow physicians to predict heart attacks and provide life-saving preventive care.

_Reith et al., 2018_), or a positive remodeling (_Varnava et al., 2002_). Features of coronary plaque vulnerability may also be assessed with different imaging modalities, such as coronary computed tomography angiography, and include also low-attenuation plaques and a higher plaque burden (_Conte et al., 2017_). These features may predict the incidence of future major cardiac events (_Conte et al., 2017_; _Prati et al., 2020_), if such high-risk plaques are not timely recognized and sealed (_Dettori et al., 2020_).

From a mechanistic perspective, plaque rupture originates from an excessive stress concentration on the fibrous cap, which trespasses the tensile strength of the material and ultimately causes material failure and consequently plaque rupture. The fibrous cap yields, in this view, a protective effect, by avoiding the exposure of prothrombotic material present in the necrotic lipid core. Thus, it is not surprising that minimal FCT has been considered the most established parameter of plaque vulnerability (_Kato et al., 2012_; _Kubo et al., 2013_; _Reith et al., 2014_; _Uemura et al., 2012_): in fact, the thickness of the fibrous cap seems a good surrogate parameter of its ability to resist stresses. Still, evaluating solely the fibrous cap addresses only the main stabilizing factor in plaque biomechanics, leaving the destabilizing factors almost completely ignored. However, plaque rupture occurs if the stress within coronary lesions exceeds the protection exerted by the fibrous cap. Therefore, it is of utmost importance to also calculate the stress within the plaque as the disrupting force within the lesion and not only focus on the protective effects of the fibrous cap. However, very little is known about the determinants of the disrupting force pushing coronary plaques to rupture. In addition, the spatial relationship between rupture and the balance of stabilizing and destabilizing forces in the plaque remains largely unexplored: in other words, whether plaque rupture occurs at the site with the lowest FCT, as point of least resistance, or rather in sites of highest stress concentration, as point of maximal disruption is unclear.

It is self-evident that such research questions cannot be addressed by 'standard' plaque imaging analysis. Therefore, we chose to develop a morpho-metric finite-element analysis to determine the influence of plaque morphology on the biomechanics of coronary lesions in patients and investigate the relationship between this calculated morpho-mechanic stress and plaque rupture.

As a source of patient-specific, high-quality images of coronary plaques for this _proof-of-concept_ study, we selected optical coherence tomography (OCT), an established intravascular imaging technique which, due to a supreme resolution (up to 10–20 μm), is able to depict peri-luminal structures with high accuracy (_Burgmaier et al., 2014_). Based on the images obtained, we generated patient-specific reconstructions of coronary plaques, which we used as a basis to analyze stress concentration as a possible predictor of plaque rupture.

# Results

## Population and lesion characteristics

Of the 20 patients with type 2 diabetes enrolled in this study, 10 underwent coronary angiography due to acute coronary syndrome and 10 due to chronic coronary syndrome. All selected patients with acute coronary syndromes showed plaque rupture as the morphological correlate. Patients with (n = 10) and without (n = 10) plaque rupture did not differ with respect to their clinical characteristics, apart from a worse glycemic control in patients with plaque rupture (HbA1c: 7.4 ± 1.4 vs. 6.1 ± 0.5, p=0.026). As expected, ruptured lesions presented a lower FCT (minimal FCT: 49 ± 10 μm vs. 97 ± 15 μm, p<0.001; mean FCT: 94 ± 17 μm vs. 133 ± 12 μm, p=0.006), a more extensive necrotic lipid core (lipid arc: 178 ± 39° vs. 110 ± 8°, p=0.001; lipid volume index: 9876 ± 3088 vs. 3853 ± 1294, p=0.011), and a higher incidence of thin-capped fibroatheromas (70% vs. 0%, p=0.003) compared to non-ruptured lesions.

Patients and lesions characteristics are reported in *Table 1*.

## Stress analysis in ruptured and non-ruptured plaques

In ruptured plaques, the maximal stress within the fibrous cap (peak cap stress) was significantly higher than in non-ruptured ones (174 ± 67 vs. 52 ± 42 kPa, p<0.001). Furthermore, also the maximal stress within the whole plaque (maximal plaque stress) was more than fourfold higher in ruptured plaques compared to stable ones (399 ± 233 vs. 90 ± 95 kPa, p=0.001). Exemplary images of stress

**Table 1.** Patients and lesions characteristics.

Abbreviations: BMI = body mass index, FCT = fibrous cap thickness, TCFA = thin-capped fibroatheroma.

|  | Non-ruptured | Ruptured | p |
|---|---|---|---|
|  | n = 10 | n = 10 |  |
| Clinical characteristics |  |  |  |
| Male sex (n, %) | 8 (80%) | 8 (80%) | 1.000 |
| Age (years) | 68 ± 7 | 70 ± 10 | 0.689 |
| Hypertension (n, %) | 10 (100%) | 9 (90%) | 0.305 |
| Hyperlipidemia (n, %) | 7 (70%) | 6 (60%) | 0.639 |
| Nicotine use (n, %) | 3 (30%) | 4 (40%) | 0.639 |
| BMI (kg/m$^2$) | 33 ± 5 | 30 ± 7 | 0.230 |
| Total cholesterol (mg/dl) | 172 ± 43 | 165 ± 33 | 0.663 |
| LDL-c (mg/dl) | 106 ± 41 | 101 ± 32 | 0.724 |
| HDL-c (mg/dl) | 39 ± 6 | 39 ± 6 | 0.972 |
| Triglycerides (mg/dl) | 161 ± 62 | 180 ± 108 | 0.640 |
| HbA1c (%) | 6.1 ± 0.5 | 7.4 ± 1.4 | 0.026 |
| hsCRP | 5.0 ± 2.3 | 26.7 ± 46.9 | 0.175 |
| Aspirine therapy (n, %) | 10 (100%) | 9 (90%) | 0.305 |
| Statine therapy (n, %) | 6 (60%) | 6 (60%) | 1.000 |
| Lesion characteristics |  |  |  |
| Minimal FCT (μm) | 97 ± 15 | 49 ± 10 | <0.001 |
| Mean FCT (μm) | 133 ± 12 | 94 ± 17 | 0.006 |
| Maximal lipid arc (°) | 110 ± 8 | 178 ± 39 | 0.001 |
| Lipid volume arc (mm*°) | 3853 ± 1294 | 9876 ± 3088 | 0.011 |
| Presence of TCFA (n, %) | 0 (0%) | 7 (70%) | 0.003 |
| Presence of macrophages (n, %) | 4 (40%) | 6 (60%) | 0.371 |
| Presence of spotty calcifications (n, %) | 7 (70%) | 8 (80%) | 0.527 |

concentrations on the fibrous cap and overall in plaques, as well as a box-plot depicting stress concentrations in lesions with and without rupture are shown in *Figure 1*. Variation of stress distribution in dependence of various model assumptions are reported in the Supplementary Results in Appendix 1.

## Spatial correlation between stress concentrations and rupture site in OCT images

After documenting much greater stress in ruptured compared with stable plaques, we assessed the effects of these morphology-driven peaks of intra-plaque stress on the lesion and if points of highest stress co-localize with plaque rupture within coronary lesions. Thus, we aimed to analyze the spatial correlation between the rupture point, as visualized in OCT images, and the point of maximal stress concentration, in order to further support the mechanistic role of stress concentrations in the fibrous cap in the genesis of plaque rupture. Only 50% of the plaque ruptures were in the shoulder region. The angle between the peak cap stress and the detectable rupture site was very low at 21.8 ± 30.3°;

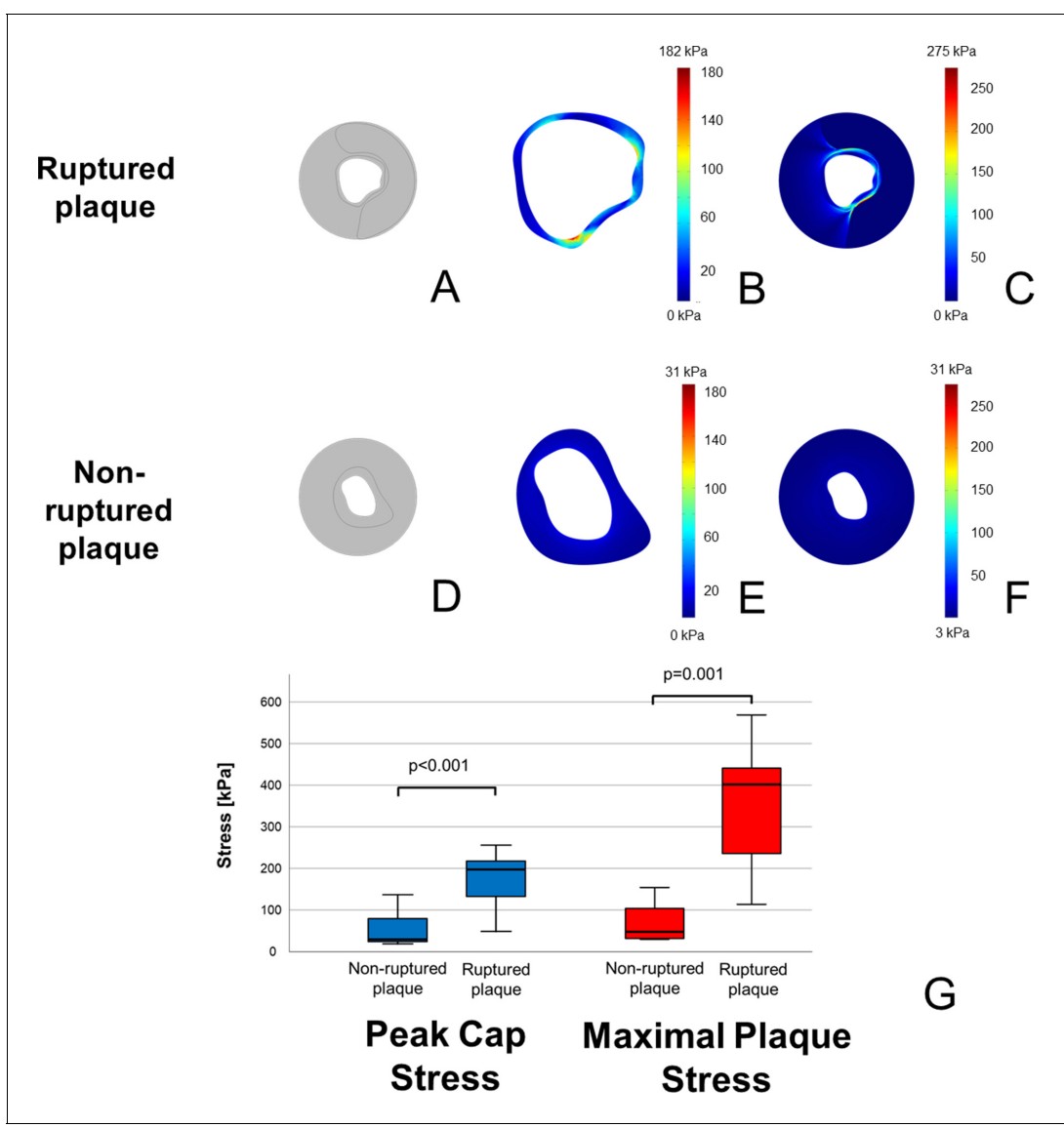

**Figure 1.** Higher stress concentrations in ruptured plaques than in non-ruptured coronary plaques. In (A, D), model reconstructions for ruptured and non-ruptured plaques are shown. In both fibrous cap (B) vs. (E) and vessel wall (C) vs. (F), ruptured plaques present higher stress concentrations than non-ruptured ones. Color-code scale is the same for both reconstructions, and maximal stress is shown at the upper edge of the scale. In (G) is shown a box-plot demonstrating the different stress concentrations both on the fibrous cap (in blue) and in the whole plaque (in red).

the angle between the maximal plaque stress and the detectable rupture site was even lower (20.7 ± 23.7˚). Overall, 50% of plaque ruptures occurred within 10˚ of the peak cap stress and of the maximal plaque stress, suggesting that plaque rupture occurs in the very proximity of highest stress concentrations. A graphical presentation of spatial correlation between the sites presenting the peak cap stress and the maximal plaque stress in finite-element analysis and the rupture site in OCT pullback is shown in *Figure 2*.

## Diagnostic value of stress distribution in predicting plaque rupture

In order to assess the diagnostic efficacy of finite-element analyses in predicting plaque rupture, we performed receiver operating characteristic (ROC) analysis for both peak cap stress and maximal plaque stress.

In our cohort, we could demonstrate that both peak cap stress (area under the curve [AUC] 0.940) and maximal plaque stress (AUC 0.950) predict plaque rupture with excellent accuracy. Optimal cut-off for prediction of plaque rupture were 150.5 kPa (Sensitivity 90%; Specificity 80%) for

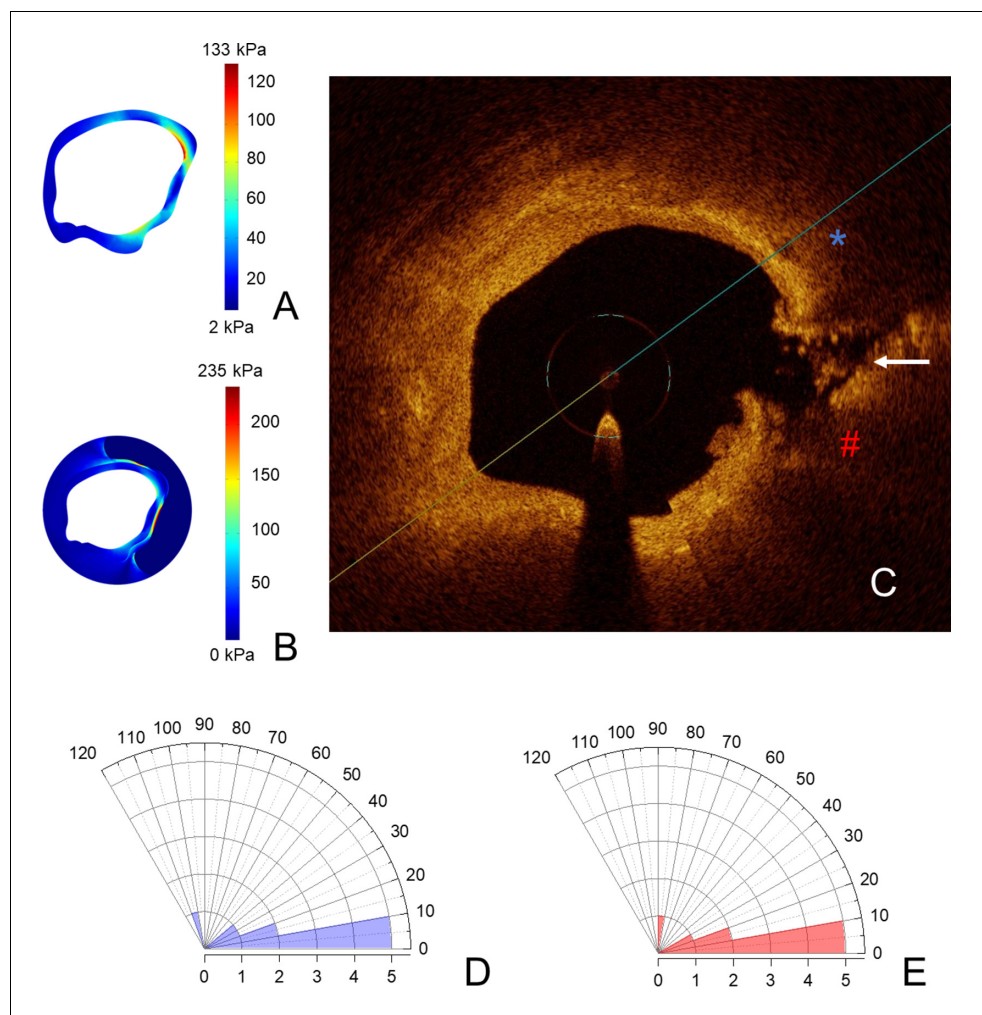

**Figure 2.** Coronary plaques rupture in the proximity of stress concentrations. In (**A, B**), stress concentrations on the fibrous cap and on the vessel wall in the same coronary segment; in (**C**), the points of maximal stress concentration on the fibrous cap (blue asterisk) and on the vessel wall (red hash) are reported, in the OCT-frame immediately following the one used for reconstruction shown. Here, the rupture is marked with a white arrow. In (**D, E**), polar histograms showing distribution of angles between rupture and maximal stress on fibrous cap (**D**) and on the vessel wall (**E**). Angles higher than 120˚ have not been detected and for this reason are not shown in the polar graphs.

peak cap stress and 169.5 kPa (Sensitivity 90%; Specificity 90%) for maximal plaque stress. ROC curves for the prediction of plaque rupture are shown in *Figure 3*.

To assess the possible additional value of finite-elements analysis compared to known features of plaque vulnerability, we performed ROC analysis for prediction of plaque rupture. Minimal FCT (AUC 0.630) and mean FCT (AUC 0.630) presented sufficient diagnostic accuracy; lipid volume index (AUC 0.870) showed a very good diagnostic accuracy in predicting plaque rupture. Both peak cap stress and maximal plaque stress presented a significantly superior diagnostic efficiency in predicting plaque rupture compared to mean FCT (p=0.027 vs. peak cap stress; p=0.036 vs. maximal plaque stress), minimal FCT (p vs. peak cap stress: 0.027; p vs. maximal plaque stress = 0.036) and extent of macrophage infiltration (p=0.003 vs. peak cap stress; p=0.001 vs. maximal plaque stress). Furthermore, diagnostic efficiency of both peak cap stress and maximal plaque stress was numerically superior (albeit non-significant) compared to that of lipid volume index (p=0.409 vs. peak cap stress and p=0.386 vs. mean plaque stress).

We combined OCT-derived morphologic features of plaque vulnerability by calculating two previously validated scores, shown in the CLIMA study (*Prati et al., 2020*) and in a publication by *Burgmaier et al., 2014*. Both scores could predict plaque rupture with very good to excellent diagnostic efficiency (AUC for CLIMA score: 0.870; AUC for Burgmaier score: 0.900), which however remained numerically lower than results of our morpho-mechanic analysis. ROC curves, including comparison with results of finite-element analyses, are shown in *Figure 4*.

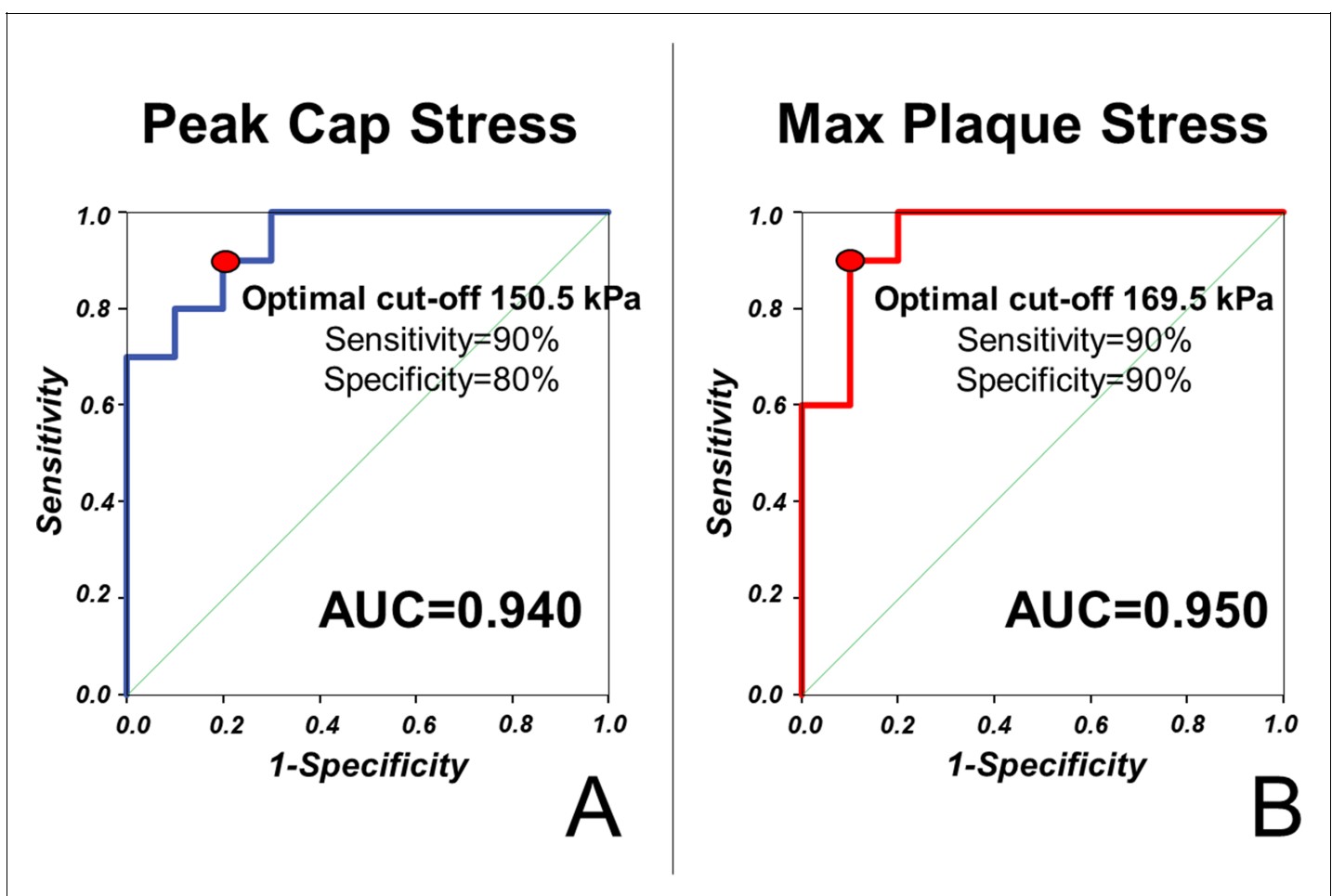

**Figure 3.** Stress concentration predicts plaque rupture with excellent diagnostic efficiency. ROC curves for the prediction of plaque rupture are shown for maximal stress in the fibrous cap (**A**) and in the vessel wall (**B**) of simulated vessels.

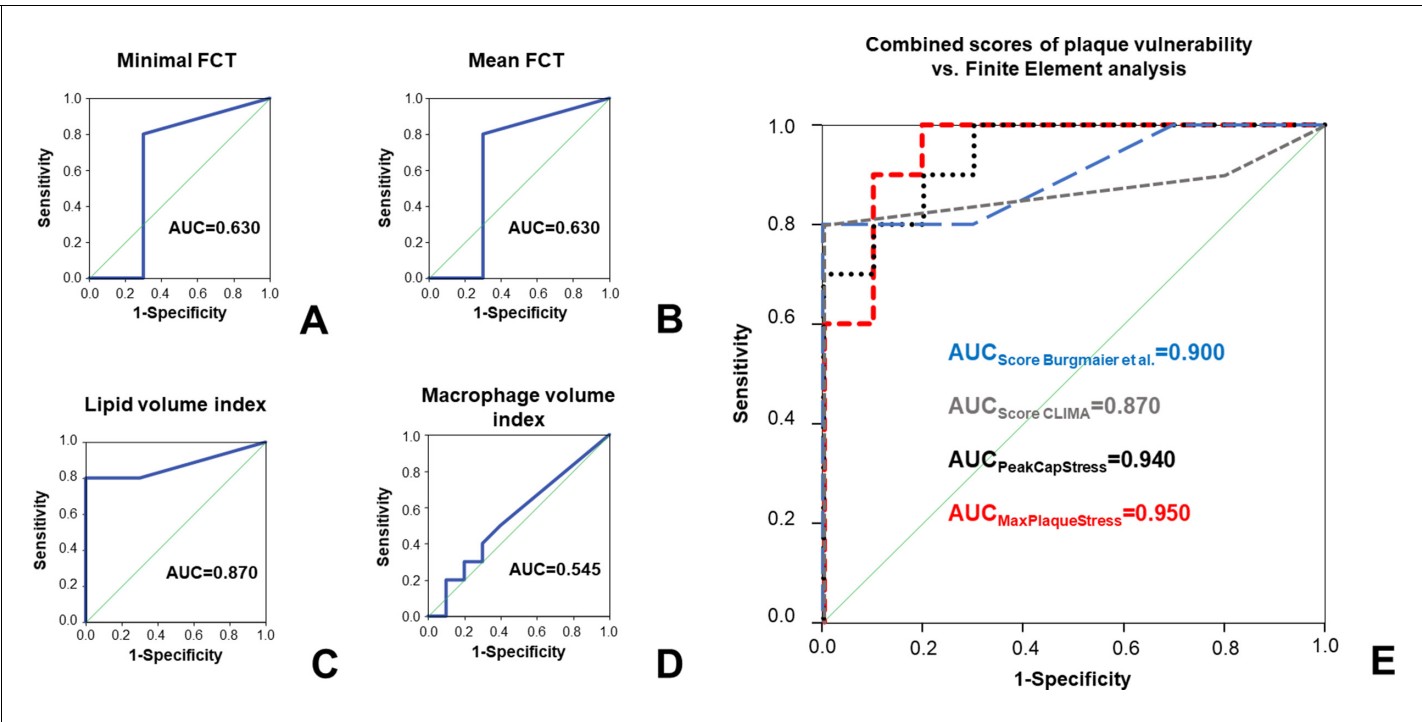

**Figure 4.** Comparison of stress concentration to 'classical' features of plaque vulnerability in prediction of plaque rupture. ROC curves for the prediction of plaque rupture for minimal FCT (A), mean FCT (B), lipid volume index (C), and macrophage volume index (D). In (E), ROC curves for two established OCT scores combining different features of plaque vulnerability are depicted (respectively: scores calculated according to *Burgmaier et al., 2014* in blue and according to the CLIMA study (*Prati et al., 2020* in grey) and compared to the ROC curves for peak cap stress (PCS, black dotted line) or maximal plaque stress (MPS, red dashed line).

## Discussion

The main findings of our work are:

- OCT-based morpho-mechanic finite-element analysis is a feasible tool to obtain patient-specific assessment of biomechanics in coronary lesions;
- our model detects significantly greater stress concentrations (more than threefold for peak cap stress and more than fourfold for maximal plaque stress) in ruptured plaques compared to non-ruptured plaques and predicts plaque rupture with excellent diagnostic efficiency;
- plaque rupture arises in the proximity of stress concentration areas;
- the diagnostic efficiency of our model was superior to established morphologic features of plaque vulnerability, as for instance the thickness of the fibrous cap, the extent of the necrotic lipid core, and the extent of plaque macrophage infiltration.

Current research on coronary plaque vulnerability mainly focused on morphological features of vulnerable plaque. The main morphologic characteristic that emerged from previous studies is FCT (*Kato et al., 2012*; *Kubo et al., 2013*; *Reith et al., 2014*; *Uemura et al., 2012*), which has been constantly related to a higher incidence of plaque rupture. This is not surprising, considering its role as the last barrier resisting the stresses exerted on the atheroma and thus preventing plaque rupture. Measuring FCT, in this perspective, is indirectly assessing the mechanical resistance – and thus the protection – of the coronary plaque to rupture. But the decrease of this protection in cases of a low FCT cannot completely determine plaque rupture, which arises from the difference between protection exerted by the fibrous cap and its disrupting forces. Data on these disrupting forces as the *primum movens* of plaque rupture are still very scarce (*Loree et al., 1992*; *Cheng et al., 1993*; *Finet et al., 2004*). Therefore, in this proof-of-concept study, we aimed to assess stress concentrations on and determined by coronary plaques in vivo and analyzed their relationship with plaque rupture. To perform this analysis, we developed a morpho-mechanic model reconstructed from in vivo OCT images.

Previous pioneering studies could show the applicability of finite-element analysis to the assessment of the biomechanics of the atherosclerotic plaque (*Loree et al., 1992*; *Cheng et al., 1993*). However, pathology-based simulations lack a direct applicability in patient care. Other studies, although based on intravascular imaging, were based on single lesions, which were analyzed on a qualitative basis (*Finet et al., 2004*; *Reith et al., 2019*). On the contrary, our study is to the best of our knowledge first to perform complex mechanistic analysis of coronary plaques based on high-definition intravascular imaging on a patient-specific level, with a high potential for direct clinical application.

## Stress concentration is closely associated to plaque rupture

First, we could in vivo detect higher stress concentrations both within the fibrous cap and within the whole plaque in ruptured plaques compared to non-ruptured plaques. This is in line with previous simulation studies based on pathology specimens (*Loree et al., 1992*; *Cheng et al., 1993*) and confirms our hypothesis of stress concentration as a central factor in the genesis of plaque rupture. As a further step in confirming this theory, we extended current knowledge by showing that plaque ruptures arise in the immediate proximity of maximal stress concentrations. This spatial coincidence between maximal stress concentrations and sites of plaque rupture confirms the causal link between plaque biomechanics on the one hand and plaque rupture and acute coronary syndromes on the other. In fact, on the basis of our data, it is tempting to speculate about the mechanistic process eventually leading to plaque rupture. Stress within the fibrous cap and overall within the plaque is concentrated in a pattern which is closely dependent on the morphology of the plaque (for instance, in dependence of the extent and morphology of the necrotic lipid core and/or of calcification). A specific force is exerted on each point of the fibrous cap; should this stress concentration trespass the ultimate tensile strength of the vessel wall – which is highly likely to happen in points of maximal stress concentration – material failure and, eventually, plaque rupture occurs.

An interesting question is, of course, the magnitude of the threshold that needs to be trespassed to cause material failure and plaque rupture. A previous pathology-based study by *Cheng et al., 1993* sets the stress threshold for plaque rupture to 300 kPa. In our study, we found 150.5 kPa in the fibrous cap and 169.5 kPa in the vessel wall to be the optimal thresholds for predicting plaque rupture. The difference between these values and the initially estimated threshold of 300 kPa may be explained through the different imaging modalities used to reconstruct the coronary plaque. In fact, in pathology specimens (on which the 300 kPa value is based on), the fixation process may cause a shrinkage of the fibrous cap with a shortening of ca. 10–20% compared to the thickness measured in vivo with OCT. This effect may be sufficient to explain the numerical difference of the 'critical' peak cap stress needed for plaque rupture, especially considering – as pointed out by Finet et al. – that the relationship between FCT and peak cap stress is exponential (*Finet et al., 2004*). Further clinical validation of these thresholds is, however, needed. Another possible explanation might be the hyperelastic behavior of certain plaque components, which has been described by some authors previously (*Yang et al., 2009*; *Cardoso et al., 2014*); such an effect is, however, only present at large displacements (>20–30% of the initial dimensions) (*Yang et al., 2009*; *Kobielarz et al., 2020*), which are way over the average displacements reported in our study. In this range, a linear equation adequately depicts the behavior of the material. Also, stress on the lesion in patients is not only derived from the static factors included in our analysis, but also dynamic flow shear stress caused by blood flow (*Bourantas et al., 2020*), which was not included in our study.

## Potential clinical utility of finite-element analysis in predicting plaque rupture

Although dynamic flow shear stress was not included in our analysis, we could also show an excellent diagnostic efficiency of our finite-element analysis in predicting plaque rupture. Furthermore, the model we developed presented a clearly superior diagnostic efficiency compared with accepted parameters of plaque vulnerability as FCT (*Kato et al., 2012*; *Kubo et al., 2013*; *Reith et al., 2014*; *Uemura et al., 2012*) or plaque macrophage infiltration (*Kato et al., 2012*; *Reith et al., 2014*; *Uemura et al., 2012*) and a numerically superior efficiency when compared to lipid volume index (*Kato et al., 2012*) in our cohort. The diagnostic efficiency of our model even presented a comparable efficiency when compared with the combination of lesion morphologies including FCT, plaque

macrophage infiltration, and lipid volume index. In the light of these findings, it may be tempting to speculate about the strong link existing between stress concentrations and plaque rupture, which may explain the excellent predictive value of our model and may pave the way for widespread use of morpho-mechanical analysis in the clinical routine to detect vulnerable plaques.

### Limitations

To the best of our knowledge, although we are first to develop an OCT-based finite-element model allowing patient-specific analysis of plaque biomechanics, several limitations have to be acknowledged. First of all, we faithfully reproduced plaque morphology in a 2D-based reconstruction; on the other hand, we are not taking into account several phenomena that may influence stress concentration, as for instance the longitudinal structure of the plaque or the longitudinal profile of the stenosis and flow shear stress – the relevance of these factors needs to be addressed in future studies.

Furthermore, in spite of the excellent resolution of OCT in the near field, we cannot exclude imprecisions in the segmentation due to limited light penetration to the deeper vessel wall. Limited tissue penetration, in fact, does not allow to assess sites with positive remodeling, which may yield higher stresses due to accelerated plaque growth and therefore cause a higher rupture risk. This needs to be assessed through different imaging modalities, such as IVUS or coronary computed tomography. Moreover, dedicated computational techniques in order to reconstruct the deep structure of the lipid core (*Kok et al., 2016*) have not been employed, in order to simplify plaque reconstruction; moderate imprecisions in the deep contour of the lipid core cannot therefore be excluded.

For our study, we employed a linear elastic model; other authors, though, suggest an hyperelastic behavior of the vessel wall, particularly for displacements > 20% of the initial dimensions (*Yang et al., 2009*; *Kobielarz et al., 2020*). In spite of an average displacement 'small enough' (10%) to justify linearity, we cannot exclude underestimation of stress concentrations for very few plaques with larger displacements.

To the best of our knowledge, although being the first to employ a morpho-metric approach in assessing plaque vulnerability on in vivo intravascular imaging, this pilot study still includes a low number of patients chosen among patients with and without plaque rupture, which may reduce the reliability of exact AUC and cut-off values in our predictive models; further analyses are needed to confirm the results of our proof-of-concept analysis.

## Materials and methods

### Patient population

We retrospectively selected 20 patients with type 2 diabetes mellitus and CAD, who underwent OCT prior to percutaneous coronary intervention at the Department of Cardiology of the University Hospital of the RWTH Aachen. Clinical presentation was stable CAD without evidence of plaque rupture in the OCT pullback (n = 10) or acute coronary syndrome with plaque rupture (n = 10). Sample size calculation was performed based on previous results of calculation of mechanical stresses in histopathological samples, resulting in a minimal sample size of seven lesions per group in order to achieve $\alpha$ = 0.001 and power of 0.95; this was then arbitrarily rounded to 10 lesions per group. Informed consent of all patients was obtained prior to inclusion in the study. The study was approved by the Ethics Committee of the University Hospital of the RWTH Aachen (EK 071/11 and EK 277/12) and is in accordance with the declaration of Helsinki on ethical principles for medical research involving human subjects.

### OCT image acquisition and analysis

The acquisition of OCT pullbacks was performed as previously described in the literature (*Tearney et al., 2012*; *Milzi et al., 2017*). In brief, OCT images were acquired prior to coronary intervention using a frequency domain OCT C7XR system and the DragonFly catheter (St. Jude Medical Systems; Lightlab Imaging Inc, Westford, MA). Blood removal was obtained by the injection of 14 ml contrast dye (iodixanol) at a flow rate of 4 ml/s through the guiding catheter. Image acquisition was obtained with automated pull-back rate of 20 mm/s.

Analysis of plaque morphology was performed as previously described (*Tearney et al., 2012*; *Milzi et al., 2017*). In particular, as widely employed in clinical practice and in previous intravascular imaging studies, FCT is measured at different sites (conventionally 3) per frame, with an analysis performed on different frames in 0.1–0.2 mm intervals. Usually, the rupture is localized in a single point or in a localized area of the cap, not impeding measurement even in the frame(s) with evident rupture, though in slightly different sites.

In order to combine different features of plaque vulnerability for comparison with results of morpho-mechanical analysis, we calculated two different, established scores. The score validated in a publication from Burgmaier et al. (following: Burgmaier Score) is calculated as −2.401 + 1.568 * (insert one if macrophages present; else 0) + 2.639 * (medium lipid arc in multiples of 90°) + 0.255 * (lipid plaque length in mm) − 0.738 * (minimal FCT in multiples of 10 µm), as previously described (*Burgmaier et al., 2014*). The score based on the CLIMA study (following: CLIMA score) attributed one point each to the presence of MLA < 3.5 mm$^2$, FCT < 75 µm, lipid arc circumferential extension > 180°, and presence of OCT-defined macrophages (*Prati et al., 2020*).

## Manual segmentation and reconstruction

In order to obtain a patient-specific model of the coronary plaque, we selected a single frame from every OCT pullback. In ruptured plaques, the frame immediately preceding the site of the rupture was chosen; this was based on the consideration that, due to rupture of the fibrous cap and to the present artifacts (caused for instance by thrombotic material), exact reconstruction of the rupture site may be inaccurate. For stable lesions, the frame with the minimal lumen area was selected; this was based on the need to select in a uniform way the site with the most advanced plaque development, assuming only a negative remodeling. After image selection, the operator (AM) was blinded to clinical presentation, as each frame was marked with a random ID. The selected image was then scaled 10:1 previous to segmentation of the different components of the atherosclerotic plaque, which was manually performed using commercial software (AutoCAD 2017, AutoDesk INC, San Rafael, CA). Plaque composition was analyzed according to the Consensus standards (*Tearney et al., 2012*). Specifically for segmentation, we first delineated the vessel lumen, defined as the signal-poor region centrally located in the OCT image. We then proceeded to tracing the boundaries of the fibrous cap, which was defined as the signal-rich region surrounding the lumen; for clarity's sake, this area will be denominated fibrous cap also when overlying a calcific or fibrocalcific plaque. When present, we segmented the necrotic lipid core, which was identified as a signal-poor region with poorly delineated borders, a fast signal drop-off, and little or no signal backscattering. When present, calcifications have also been segmented; as calcifications we considered signal-poor or heterogeneous regions with a sharply delineated border. In case of non-detectable borders of each of the segmented components due to artifacts or to the limited penetration of light, we delineated contours with the automatic interpolation function of the software. We conventionally shaped the coronary vessel as a cylinder and set its external diameter to 4 mm. A sample reconstruction is shown in *Figure 5*.

In order to obtain a three-dimensional model of the vessel, we performed a graphical extrusion of the segmented contours over a length of 10 mm.

## Finite-element analysis

The modeled vessel was then imported in commercial software to perform finite-element analysis (COMSOL Multiphysics 5.0, Stockholm, Sweden). A solid mechanics physics was chosen, and the used mechanical properties for the different components of the plaque were extrapolated from previous literature (*Loree et al., 1992*; *Cheng et al., 1993*; *Finet et al., 2004*; *Reith et al., 2019*) and are reported in *Table 2*.

Nevertheless, in the literature, there is no consensus regarding mechanical properties of the atherosclerotic plaque. This is a consequence of the very limited number of samples used for mechanical testing, of the need for pre-treatment of pathology samples (which may potentially alter their mechanical properties), and of the intrinsic difficulty of the measurement of some properties (specifically, the Poisson's ratio, which is only indirectly measurable). In particular, some authors hypothesized an incompressible behavior of the vessel wall, which would lead to a Poisson's ratio of 0.48 for this component (instead of 0.27 as used in our simulations) (*Yang et al., 2009*). Moreover, very

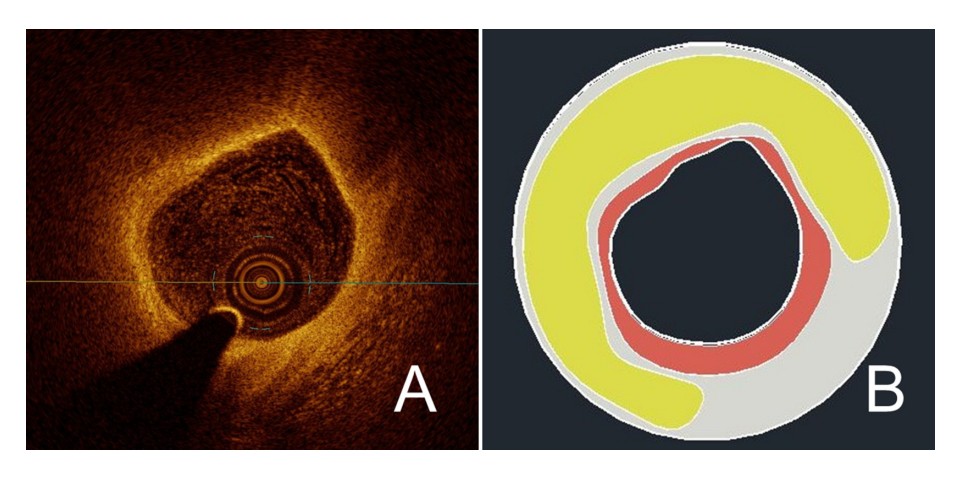

**Figure 5.** Exemplary reconstruction of a coronary plaque. Using manual segmentation, a reconstruction of a vessel segment was performed based on OCT images. In (A), the source image from the OCT pullback shows a vulnerable plaque with extensive lipid core and thin fibrous cap. In (B), the result after manual segmentation of the plaque components is shown; here, fibrous cap is in red, lipid core in yellow, and the rest of the vessel wall in light grey.

The online version of this article includes the following figure supplement(s) for figure 5:

**Figure supplement 1.** Graphical representation of the finite-element model.

**Figure supplement 2.** Effects of constraints on the terminal surfaces on stress distribution and displacement.

**Figure supplement 3.** Stress analysis throughout the simulated vessel segment.

different values of stiffness of calcifications have been detected and employed in previous studies, ranging from about 10 MPa for mildly calcified tissues to even 17–25 GPa emulating the properties of bone tissue (*Loree et al., 1992*; *Cheng et al., 1993*; *Finet et al., 2004*; *Kobielarz et al., 2020*; *Holzapfel et al., 2005*; *Barrett et al., 2019*; *Wong et al., 2012*; *Cahalane et al., 2018*). Specifically, the use of smaller Young's moduli is justified by the inhomogeneity of macrocalcifications, which may be only partly constituted by crystalline calcium *Wong et al., 2012*; this could be associated, in previous studies, to the density of calcifications in computed tomography (*Cahalane et al., 2018*). Though, as calcium density is not defined in OCT, we preferred to use 10 GPa, as the Young's modulus of crystalline calcium, as calcification's stiffness. Nevertheless, in order to exclude a relevant impact of these assumptions on our stress analysis, we assessed stress distribution in the presence of different assumptions. If not differently specified, the data presented in the rest of the manuscript derive from the model using constants shown in *Table 2*.

A pressure of 130 mmHg (= 17 kPa) on the luminal side was applied as external load and a simulation of the structural stress distribution in response to the load was performed. The round outer surface of the vessel was kept as fixed constraint. A suitable mesh was chosen per each simulated vessel, ranging from 'fine' to 'very fine', in order to keep computational times reasonable. Average mesh properties are reported in *Supplementary file 1*. Then, we analyzed the stress distribution as von Mises stress in a bi-dimensional cross-section of the vessel normal to the vessel axis; in order to avoid edge effects, we used cross-sections at a distance of 5 mm from each end. A graphical representation of the model is included in *Figure 5—figure supplement 1*. The stress intensity on the

**Table 2.** Mechanical properties of the different plaque components.

|  | Poisson's ratio (ν) | Young's modulus |
|---|---|---|
| Fibrous cap | 0.27 | 244 kPa |
| Necrotic lipid core | 0.48 | 1 kPa |
| Calcification | 0.30 | 10 GPa |
| Vessel wall | 0.27 | 800 kPa |

fibrous cap was graphically shown on a blue-red color scale. The highest von Mises stress in the fibrous cap was defined as 'Peak Cap Stress'; the highest von Mises stress in the vessel wall was defined as 'Maximal Plaque Stress'.

The finite-element analysis was performed in a dedicated core lab from an operator (EDL) blinded to the clinical presentation of the patients.

To exclude excess interobserver variability in manual segmentation, a different experienced OCT operator (RD) redraw the analyzed structures in a randomly selected 20% of the plaques in each group. Based on these segmentations, we run finite-element analysis.

## Spatial correlation analysis between stress concentration and rupture site

Rupture site was identified in OCT images as a clear, visible continuity interruption in the fibrous cap. The rupture site was marked from an operator (AM) blinded to the results of stress distribution. In case of non-punctual ruptures of the fibrous cap, an arbitrary middle point was used for calculation purposes. Then, results of stress analysis for both the fibrous cap and the overall vessel wall were reported on the OCT-image of the rupture site, correlating them to anatomic landmarks (morphology of lumen; morphology of the fibrous cap; presence of calcifications, lipid deposits, cholesterol crystals, or macrophage accumulations). The angular distance between the rupture site and the sites where the simulation highlighted maximal stress concentrations was noted. Graphical representation was obtained with polar angle histograms generated with Origin (OriginLab Corp, Northampton, MA).

## Statistical analysis

Continuous variables were reported as mean ± standard deviation, categorical as count (percentage). Distributions of continuous variables were compared using t-test; for comparing categorical values, Fisher's exact test was used for comparing distribution of categorical variables. To compare results of simulations with different assumptions regarding material properties, paired-samples t-test was used. Furthermore, in order to validate the finite-element simulation, we correlated results obtained in different models and after segmentation through different operators using one-way random effects model; results were expressed as intraclass correlation coefficient. We performed ROC analysis to validate diagnostic value of the results of simulation models as well as 'classical' features of plaque vulnerability (FCT, lipid volume index, extent of plaque macrophages infiltration) in predicting plaque rupture. Values with the highest Youden index were identified as optimal cutoff values; in case of equal Youden index between two or more data points, we selected one based on clinical judgment. In order to evaluate the value of a combination of OCT-derived morphologic parameters to predict plaque rupture, we performed multivariable logistic regression including minimal or mean FCT, lipid volume index, and macrophage volume index. Lipid volume index was calculated as the product of mean lipid angle and lipid length, as defined in previous works (*Kato et al., 2012*; *Kubo et al., 2013*; *Reith et al., 2014*; *Uemura et al., 2012*; *Ehara et al., 2004*; *Reith et al., 2018*; *Tearney et al., 2012*; *Milzi et al., 2017*). For calculation purposes, non-defined parameters such as FCT in case of calcified plaques or lipid volume index in non-lipidic plaques were set to zero. Then, ROC analysis was performed based on the predictive values of this multiple regression model. A classification of the diagnostic efficiency according to the values of the area under the curve (AUC) was used as described elsewhere (*Šimundić, 2009*). In order to compare diagnostic efficiency of results of morpho-mechanic analysis with combined features of coronary plaque vulnerability, we calculated the Burgmaier and CLIMA scores as previously described. Comparison of the diagnostic efficiency among different ROC curves was performed with the DeLong test, as previously described (*DeLong et al., 1988*). All statistical analyses were performed with SPSS software (v. 26.0, IBM Corp., Armonk, NY). Statistical significance was awarded for p<0.05.

## Conclusion

In our proof-of-concept study, we demonstrate that OCT-based finite-element analysis is a feasible tool to determine plaque biomechanics, which in turn may predict plaque rupture in patients. Whereas the minimal fibrous cap thickness protects the plaque from its rupture, our data highlight the importance of morpho-mechanic analysis assessing the disrupting effects of plaque stress. These

data need, however, to be verified in larger populations. This new method may offer valuable insights on the interplay between various plaque components in the determination of the net vulnerability of a plaque, bringing stress concentrations – the disrupting force of plaque rupture – back into clinical practice.

# Acknowledgements

EDLs postdoctoral position at Karlsruhe Institute of Technology is funded via a research fellowship of the Alexander-von-Humboldt Foundation.

# Additional information

### Funding

| Funder | Grant reference number | Author |
| --- | --- | --- |
| Alexander von Humboldt-Stiftung | Postdoctoral Research Fellowship | Enrico Domenico Lemma |

The funders had no role in study design, data collection and interpretation, or the decision to submit the work for publication.

### Author contributions

Andrea Milzi, Conceptualization, Data curation, Formal analysis, Validation, Investigation, Visualization, Methodology, Writing - original draft; Enrico Domenico Lemma, Conceptualization, Data curation, Validation, Investigation, Visualization, Methodology, Writing - original draft; Rosalia Dettori, Conceptualization, Formal analysis, Investigation, Methodology, Writing - review and editing; Kathrin Burgmaier, Data curation, Formal analysis, Writing - review and editing; Nikolaus Marx, Sebastian Reith, Conceptualization, Supervision, Methodology, Writing - review and editing; Mathias Burgmaier, Conceptualization, Supervision, Investigation, Methodology, Project administration, Writing - review and editing

### Author ORCIDs

Andrea Milzi https://orcid.org/0000-0001-7580-8029

### Ethics

Human subjects: The study was approved by the Ethics Committee of the University Hospital of the RWTH Aachen (EK 071/11 and EK 277/12) and is in accordance with the declaration of Helsinki on ethical principles for medical research involving human subjects.Informed consent of all patients was obtained prior to inclusion in the study.

### Decision letter and Author response

Decision letter https://doi.org/10.7554/eLife.64020.sa1
Author response https://doi.org/10.7554/eLife.64020.sa2

# Additional files

### Supplementary files

• Supplementary file 1. Characteristics of the mesh. Maximum and minimum element size, respectively, limits how big and how small each mesh element can be. Maximum element growth rate limits the size difference of two adjacent mesh elements. Curvature factor limits how big a mesh element can be along a curved boundary. Resolution of narrow regions controls the number of layers of mesh elements in narrow regions.

• Transparent reporting form

## Data availability

Relevant source data are included in the paper: - mechanical properties assumed for analysis are included in Table 1 - assumptions for FEA analysis are clearly reported in Methods and Supplementary material (Supplementary Text, Suppl. Figures 1-3) - data analysis was performed with standard SPSS package with no use of user-compiled code - parameters entering multivariable analysis and selection criteria are stated in the "statistical analysis" subsection.

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

# Appendix 1

## Supplementary material
### Supplementary results
Validation of the finite-element analysis model

In order to test some possible limitations of our models, we performed several analyses and controls.

Firstly, we analyzed the magnitude of the displacements of the simulated components of the vessel wall. Average displacement was 0.39 ± 0.45 mm. These values are compatible with the biological dimensions of the considered vessel. Due to this average displacement of approximately 10% of the original dimensions of the considered vessel (4 mm), we assumed the linear elasticity of the vessel wall for further analysis, as previous publication showed a hyperelastic behavior starting from much larger deformations (>20–30%) (*Yang et al., 2009*; *Kobielarz et al., 2020*).

Secondly, in order to analyze the effects of considering vessel ends as fixed constraints, we performed finite-element analysis both with and without these boundary conditions. We could detect no relevant difference between these two scenarios in the exemplarily analyzed vessels (respectively, intraclass correlation coefficients 0.998 for maximal stress in the fibrous cap, 1.000 for maximal stress in the whole plaque, and 1.000 for maximal displacement). An example of the stress distributions and displacements with and without constraint is shown in *Figure 5—figure supplement 2*. As the calculated difference was minimal (<5%), for further analyses we employed the model without constraints on the vessel ends.

Then, in order to assess the magnitude of edge effects, we analyzed stresses in 2 mm intervals along the vessel axis. Exemplary results are shown in *Figure 5—figure supplement 3*. As expected, differences were minimal (<5%) between following sections and approximately symmetrical with respect to the vessel ends, which led us to consider the cross-sections with a distance of 5 mm from each end for further analyses.

As biomechanical properties of atherosclerotic coronary vessel are not unanimously accepted in the literature due to the scarcity of data and the difficulty of measurement, we aimed to test the robustness of our model by assessing the variations of stress concentrations in the presence of different assumptions. By accepting the hypothesis of incompressible materials advanced in some previous works (*Yang et al., 2009*) and thus setting a Poisson's ratio of 0.48, no significant difference in MPS could be detected in the overall population (MPS: 244.5 ± 234.4 kPa with $\nu$ = 0.27 vs. 240.4 ± 248.2 kPa with $\nu$ = 0.48; p=0.433; intraclass correlation coefficient 0.996). This reflects previous findings that variations of the Poisson's ratio does not play a major role in plaque biomechanics (*Baldewsing et al., 2004*). Due to the minimal impact of this assumption on results of our analysis, we employed the Poisson's ratios indicated in *Table 2* for further analysis.

Similarly, as properties of calcification may vary due to the density of calcium in the intravascular calcification (*Loree et al., 1992*; *Cheng et al., 1993*; *Finet et al., 2004*; *Kobielarz et al., 2020*; *Holzapfel et al., 2005*; *Barrett et al., 2019*; *Wong et al., 2012*; *Cahalane et al., 2018*), we rerun our finite-element analysis including a much lower stiffness (1 MPa) for calcifications. In the overall population, this had no significant impact on stress concentration (MPS: 244.5 ± 234.4 kPa with $E_{Calcification}$ = 10 GPa vs. 246.1 ± 242.1 kPa with $E_{Calcification}$ = 1 MPa; p=0.753; intraclass correlation coefficient 0.996). As this effect may be due to the limited incidence of calcified plaques in our study (n = 5, 20%), we assessed the role played by a lower stiffness of calcification in stress concentration in this specific subgroup of lesions. Again, no significant difference could be detected (MPS in calcified lesions: 115.3 kPa with $E_{Calcification}$ = 10 GPa vs. 114.1 ± 137.3 kPa with $E_{Calcification}$ = 1 MPa; p=0.954). Again, in the light of the very small variations of the results, we employed the stiffness values indicated in *Table 2* for further analysis.

In order to assess the operator dependence of our methods, we rerun finite-element analysis based on segmentation performed by a second operator blinded to the first segmentation on 15% of the initial sample. We found a very good concordance with an intraclass correlation coefficient of 0.976 (p=0.003).

