## [Decision Letter]

**Acceptance summary:**

This retrospective clinical study investigated markers of biomechanical stress and plaque rupture in diabetic patients with coronary atherosclerosis. The authors modeled arterial plaque anatomy and characteristics of arterial wall stress in patients with either stable coronary artery disease (stable CAD) or acute coronary syndrome (ACS) using optical coherence tomography (OCT) imaging. The authors identified a cut-off stress value that allows the prediction plaque rupture. The results hold potential to be used for assessment of the plaque rupture risk in patients with CAD using OCT.

**Decision letter after peer review:**

Thank you for submitting your article "Coronary plaque composition influences biomechanical stress and predicts plaque rupture – a morpho-mechanic OCT analysis" for consideration by *eLife*. Your article has been reviewed by 3 peer reviewers, one of whom is a member of our Board of Reviewing Editors, and the evaluation has been overseen by a Senior Editor. The following individual involved in review of your submission has agreed to reveal their identity: José Félix Rodriguez Matas (Reviewer #2).

The reviewers have discussed the reviews with one another, and this decision letter is to help you prepare a revised submission.

Summary:

This work presents results on the finite element analysis of OCT-reconstructed plaque "3D-models" of 10 ruptured and 10 non-ruptured lesions. The manuscript is well written and structured. The results are found to be of great interest for the assessment of the plaque rupture risk in patients. The study confirms the excellent correlation between the location of the maximum stress in the arterial tissue and the location of the plaque rupture already reported by the authors in a previous study (Reith et al., Cardiovasasc Diabetol 2019; 18(1): 122). In addition, the study finds that in the ruptured lesions, the maximal stress in the fibrous cap and vessel wall were significantly higher than in the non-ruptured plaques. So, the stress concentration excellently predicts plaque rupture, underlying the importance of morpho-mechanic analysis to assess the disrupting effects of plaque stress. However, the ROC curves for the prediction of plaque rupture indicate an Optimal cut-off stress of 150kPa and 169.5kPa for the fiber cap and the plaque respectively. This threshold value is significantly lower than the threshold value of 300kPa reported in literature. The authors explain this difference through the different imaging modalities used to reconstruct the coronary plaque. However, a more plausible explanation can be found in the finite element model itself, in particular in the mechanical properties used for the analysis.

Essential revisions:

1. In the introduction the authors mention among the accepted features of plaque vulnerability: thickness of the fibrous cap, the extent of necrotic lipid core. However, recent studies also indicate the positive remodeling index, low-attenuation plaque, and plaque burden as significant markers of plaque vulnerability. I suggest to extend the literature review in this regard and to also include a recent analysis published in Eur Heart J Cardiovasc Imaging. 2017 Oct 1;18(10):1170-1178. doi: 10.1093/ehjci/jew200. PMID: 27679600.

2. Stress analysis considered an oversimplified model for the tissues. The authors make reference to rather old publications [14] 1992, [15] 1993, [16 ] 2004, [17] 2019 that do not certainly provide adequate information. The stress field, and therefore the peak stress value, will be significantly affected by the adopted mechanical model for the tissue. You are assuming the vessel wall with a Poisson's ratio of ~0.3 when in reality it is a quasi-incompressible material with a Poisson ratio ~0.5. In addition, the authors are using a value for the initial Young modulus for the calcification that is 4 orders of magnitude larger than those reported in literature. A similar observation is made for the arterial tissue, assumed as linear elastic when in reality is non-linear. Using a non-linear hyper-elastic material model for the arterial wall and the plaque will significantly affect the peak wall stress in ruptured and non-ruptured plaques and could modify the ROC curves. In conclusion, the authors should rerun the finite element analysis of all models with adequate material parameters. Please check the work by Yang et al. IEEE Trans Biomed Eng 56(10): 2420-2428 (2009), and references within to update the material parameters used in your analysis and rerun the simulations. Make sure you are using a quasi-incompressible material by imposing a Poisson's ratio of at least 0.48 in the case that a hybrid finite element formulation is not available in the software.

3. Also, the reviewers recommend including a comment regarding the choice of the frame with the minimal lumen area for stable lesions in the Section "Manual segmentation and reconstruction", just for completeness. In the case of ruptured plaques, the reviewers think that the frame immediately preceding the site of the rupture is the best choice. However, in the case of non-ruptured plaques, the choice of the frame with the minimal lumen area is the more intuitive from an engineering point of view, while from a clinical point of view it could be also interesting to assess a section with a positive remodeling, if present.

4. The 3D model was reconstructed by extruding the segmented section for a total length of 10mm. At this point, the results are equivalent to considering a plane-strain model of the plaque i.e., a 2D model. This will significantly reduce the computational cost and significantly simplify the analysis of the results. This fact is demonstrated in Suppl. Figure 3 that show the same results in sections C, D, and E, away from the end caps were boundary effects make the stress field unreliable. Finally, a convergence curve of the mesh size will be appreciated, as well as the average number of nodes and elements used for the analysis

5. Insert a comment in the limitations regarding the sample size. Twenty plaques are not a small number but maybe not sufficiently large as to completely ensure the conclusions of the manuscript. The results are very promising though.

6. Although the authors aptly analyzed the correlation between biomechanical stress and other plaque characteristics, it would be even better and more clinically applicable if the authors could combine the aforementioned factors as a single prediction tool, such as scoring system or a simplified formula to ease its application.

7. Did the authors look at the paper: "Kok AM, Speelman L, Virmani R, van der Steen AF, Gijsen FJ, Wentzel JJ. Peak cap stress calculations in coronary atherosclerotic plaques with an incomplete necrotic core geometry. Biomed Eng Online. 2016 May 4;15(1):48. doi: 10.1186/s12938-016-0162-5. PMID: 27145748; PMCID: PMC4857277."

---

## [Author Response]

Essential revisions:1. In the introduction the authors mention among the accepted features of plaque vulnerability: thickness of the fibrous cap, the extent of necrotic lipid core. However, recent studies also indicate the positive remodeling index, low-attenuation plaque, and plaque burden as significant markers of plaque vulnerability. I suggest to extend the literature review in this regard and to also include a recent analysis published in Eur Heart J Cardiovasc Imaging. 2017 Oct 1;18(10):1170-1178. doi: 10.1093/ehjci/jew200. PMID: 27679600.

The reviewer’s point is well taken. We expanded the literature review on this point, including also the important citation mentioned.

Introduction,

“The presently accepted features assessed in intravascular imaging include the thickness of the fibrous cap (FCT) [6-9], the extent of the necrotic lipid core [6], the presence of macrophages [6,8,9], microvessels [6,9], small calcifications [10,11] or a positive remodelling [Varnava et al., Circulation 2005]. Features predictive of future cardiovascular events may also be assessed with other imaging modalities, such as coronary computed tomography angiography, and include also low-attenuation plaques and a higher plaque burden [Conte et al., Eur Heart J CV Imaging 2017; Prati F et al., Eur Heart J 2019], if such high-risk plaques are not timely recognized and sealed [Dettori R et al., Cardiovasc Diabetol 2020].”

2. Stress analysis considered an oversimplified model for the tissues. The authors make reference to rather old publications [14] 1992, [15] 1993, [16 ] 2004, [17] 2019 that do not certainly provide adequate information. The stress field, and therefore the peak stress value, will be significantly affected by the adopted mechanical model for the tissue. You are assuming the vessel wall with a Poisson's ratio of ~0.3 when in reality it is a quasi-incompressible material with a Poisson ratio ~0.5. In addition, the authors are using a value for the initial Young modulus for the calcification that is 4 orders of magnitude larger than those reported in literature. A similar observation is made for the arterial tissue, assumed as linear elastic when in reality is non-linear. Using a non-linear hyper-elastic material model for the arterial wall and the plaque will significantly affect the peak wall stress in ruptured and non-ruptured plaques and could modify the ROC curves. In conclusion, the authors should rerun the finite element analysis of all models with adequate material parameters. Please check the work by Yang et al. IEEE Trans Biomed Eng 56(10): 2420-2428 (2009), and references within to update the material parameters used in your analysis and rerun the simulations. Make sure you are using a quasi-incompressible material by imposing a Poisson's ratio of at least 0.48 in the case that a hybrid finite element formulation is not available in the software.

We thank the reviewers and the editor for the opportunity to clarify this important point. We agree that the choice of input data for FEM simulations in coronary plaque analysis is a complex topic, especially regarding materials properties which are difficult to measure directly, e.g. Poisson´s ratio. Moreover, mechanical values are not measured on in vivo samples, but on animal or human specimen collected post mortem and treated for preservation, which may alter their properties. We briefly commented on it in the revised manuscript.

In order to follow the reviewer´s suggestions, we performed the required experiments by 1) accepting the hypothesis of incompressible materials (setting ν≈0.5) and 2) setting the Young´s modulus of calcification at 1 MPa. However, this did not relevantly change the results of our analysis. In order to further improve our manuscript and in order to address the reviewer´s point, we include these analyses in the revised Supplementary Results. Since these assumptions do not meet overall consensus in the literature and, most importantly, do not show a significant effect on stress concentrations, we chose to also show our initial model (ν=0.27, E_Calcification_=10GPa) in the manuscript.

Finally, the reviewers suggested to use a hyperelastic model to better simulate the mechanical behavior of the materials upon larger displacements. Though, the material properties assumed in the literature for hyperelastic behavior of the vessel wall are often extrapolated by single sample analysis and very different in the studies considered. After thoughtful consideration, we maintained a linear elastic model, because in most simulations the resulting displacement field is lower than 20% [Kobielarz M, J Mech Behav Biomed Mater. 2020] of the vessel lumen approximate diameter (average displacement: 0.39±0.45mm). Therefore, deformations can still be considered as „small enough“ to justify the linearity. We agree that for larger displacement fields, a hyperelastic model could better describe the vessel behavior and in turn increase the calculated Peak Cap Stress or Maximum Plaque Stress; this seems to be the case, though, starting from a displacement ratio of 1.2 – a condition not commonly met in our simulations. As the reviewer´s point is well taken, though, we included this in the Limitation section.

Methods, Finite elements analysis,

“Nevertheless, in the literature there is no consensus regarding mechanical properties of the atherosclerotic plaque. […] If not differently specified, the data presented in the rest of the manuscript derive from the model using constants shown in Table 2.”

Suppl. Results, Validation of the finite element analysis model,

“Firstly, we analyzed the magnitude of the displacements of the simulated components of the vessel wall. […] Again, no significant difference could be detected (MPS in calcified lesions: 115.3 kPa with ECalcification=10GPa vs. 114.1±137.3 kPa with ECalcification=1MPa; p=0.954). Again, in the light of the very small variations of the results, we employed the Poisson´s ratios indicated in Table1 for further analysis.”

Discussion, Stress concentrations…,

“Further clinical validation of these thresholds is, however, needed. Another possible explanation might be the hyperelastic behavior of certain plaque components, which has been described by some authors previously [Yang C et al., IEEE Trans Biomed Eng. 2009; Cardoso L et al., J Biomech. 2014]; such an effect is, however, only present at large displacements (>20-30% of the initial dimensions) [Yang C et al., IEEE Trans Biomed Eng. 2009; Kobielarz M, J Mech Behav Biomed Mater. 2020], which are way over the average displacements reported in our study. In this range, a linear equation adequately depicts the behavior of the material.”

Discussion, Limitations,

“For our study, we employed a linear elastic model; other authors, though, suggest an hyperelastic behavior of the vessel wall, particularly for displacements>20% of the initial dimensions [Yang C et al., IEEE Trans Biomed Eng. 2009; Kobielarz M, J Mech Behav Biomed Mater. 2020]. In spite of an average displacement “small enough” (10%) to justify linearity, we cannot exclude underestimation of stress concentrations for very few plaques with larger displacements.”

3. Also, the reviewers recommend including a comment regarding the choice of the frame with the minimal lumen area for stable lesions in the Section "Manual segmentation and reconstruction", just for completeness. In the case of ruptured plaques, the reviewers think that the frame immediately preceding the site of the rupture is the best choice. However, in the case of non-ruptured plaques, the choice of the frame with the minimal lumen area is the more intuitive from an engineering point of view, while from a clinical point of view it could be also interesting to assess a section with a positive remodeling, if present.

We thank the reviewer for pointing this out to us. We commented on the frame selection in the revised text. Regarding positive remodeling, we agree with the reviewer that these areas may be interesting in a clinical perspective; however, it must be noted that OCT, due to its limited tissue penetration, is not adequately suitable to assess positive remodeling. As this remains a limitation, we acknowledged it in the “Limitations” section of the revised manuscript.

Methods, Manual segmentation and reconstruction,

“In order to obtain a patient-specific model of the coronary plaque, we selected a single frame from every OCT-pullback. In ruptured plaques, the frame immediately preceding the site of the rupture was chosen; this was based on the consideration that, due to rupture of the fibrous cap and to the presents artefacts (caused for instance by thrombotic material), exact reconstruction of the rupture site may be largely inaccurate. For stable lesions, the frame with the minimal lumen area was selected; this was based on the need to select in a uniform way the site with the most pronounced plaque development, assuming only a negative remodeling.”

Discussion, Limitations,

“Furthermore, in spite of the excellent resolution of OCT in the near-field, we cannot exclude imprecisions in the segmentation due to limited light penetration to the deeper vessel wall. Limited tissue penetration, in fact, does not allow to assess sites with positive remodeling, which may yield higher stresses due to accelerated plaque growth and therefore cause a higher rupture risk. This needs to be assessed through different imaging modalities, such as IVUS or coronary computed tomography.”

4. The 3D model was reconstructed by extruding the segmented section for a total length of 10mm. At this point, the results are equivalent to considering a plane-strain model of the plaque i.e., a 2D model. This will significantly reduce the computational cost and significantly simplify the analysis of the results. This fact is demonstrated in Suppl. Figure 3 that show the same results in sections C, D, and E, away from the end caps were boundary effects make the stress field unreliable. Finally, a convergence curve of the mesh size will be appreciated, as well as the average number of nodes and elements used for the analysis

We agree with the reviewers about the 3D complexity of our system. We have rephrased some sentences in order to avoid possible confusion for the reader. We have also included further information on the used mesh in the revised Methods section.

Methods, Finite elements analysis,

“A suitable mesh was chosen per each simulated vessel, ranging from “fine” to “very fine”, in order to keep computational times reasonable. Average mesh properties are reported in Suppl. Table 1.”

Suppl. Table 1. Characteristics of the mesh. Maximum and minimum element size respectively limits how big and how small each mesh element can be. Maximum element growth rate limits the size difference of two adjacent mesh elements. Curvature factor limits how big a mesh element can be along a curved boundary. Resolution of narrow regions controls the number of layers of mesh elements in narrow regions.

5. Insert a comment in the limitations regarding the sample size. Twenty plaques are not a small number but maybe not sufficiently large as to completely ensure the conclusions of the manuscript. The results are very promising though.

We thank again the reviewer for the positive assessment of our work and acknowledged this limitation not only in the “Limitations” section, but also in the conclusions as required.

Conclusion,

“In our proof-of-concept study we demonstrate that OCT-based finite element analysis is a feasible tool to determine plaque biomechanics, which in turn may predict plaque rupture in patients. Whereas the minimal fibrous cap thickness protects the plaque from its rupture, our data highlight the importance of morpho-mechanic analysis assessing the disrupting effects of plaque stress. These data need, however, to be verified in larger populations.”

6. Although the authors aptly analyzed the correlation between biomechanical stress and other plaque characteristics, it would be even better and more clinically applicable if the authors could combine the aforementioned factors as a single prediction tool, such as scoring system or a simplified formula to ease its application.

The point of the reviewer is well taken. In order to simplify the comparison of the results of morphomechanic analysis with a combination of features of plaque vulnerability, we calculated two established scores depicting plaque vulnerability and compared them with the stresses computed in our model.

Methods, OCT image acquisition and analysis,

“In order to combine different features of plaque vulnerability for comparison with results of morphomechanical analysis, we calculated two different, established scores. The score validated in a publication from Burgmaier et al. (following: Burgmaier Score) is calculated as -2.401 + 1.568 * (insert 1 if macrophages present; else 0) + 2.639 * (medium lipid arc in multiples of 90°) + 0.255 * (lipid plaque length in mm) – 0.738 * (minimal FCT in multiples of 10 μm), as previously described [Burgmaier et al., Cardiovascular Diabetology 2014]. The score based on the CLIMA study (following: CLIMA score) attributed 1 point each to presence of MLA <3.5 mm2, FCT <75 µm, lipid arc circumferential extension >180°, and presence of OCT-defined macrophages [Prati F et al., Eur Heart Journal 2020].”

Methods, Statistical analysis,

“In order to compare diagnostic efficiency of results of morpho-mechanic analysis with combined features of coronary plaque vulnerability, we calculated the Burgmaier and CLIMA scores as previously described. Comparison of the diagnostic efficiency among different ROC-curves was performed with the DeLong-test, as previously described [DeLong et al., Biometrics 1988].”

Results, Diagnostic value […],

“We combined OCT-derived morphologic features of plaque vulnerability by calculating two previously validated scores, shown in a publication by Burgmaier et al. [Burgmaier et al., Cardiovascular Diabetology 2014] and in the CLIMA study [Prati F et al., Eur Heart Journal 2020]. Both scores could predict plaque rupture with very good to excellent diagnostic efficiency (AUC for Burgmaier Score: 0.900; AUC for CLIMA-score: 0.870), which however remained numerically lower than results of our morphomechanic analysis. ROC-curves, including comparison with results of finite element analyses, are shown in Figure 4.”

7. Did the authors look at the paper: "Kok AM, Speelman L, Virmani R, van der Steen AF, Gijsen FJ, Wentzel JJ. Peak cap stress calculations in coronary atherosclerotic plaques with an incomplete necrotic core geometry. Biomed Eng Online. 2016 May 4;15(1):48. doi: 10.1186/s12938-016-0162-5. PMID: 27145748; PMCID: PMC4857277."

We thank the reviewer for pointing this interesting study out to us. As we did not employ this optimization method in our study, we addressed the point as a limitation of our work.